# A Flexible and Highly Sensitive Pressure Sense Electrode Based on Cotton Pulp for Wearable Electronics

**DOI:** 10.3390/polym15092095

**Published:** 2023-04-28

**Authors:** Mengying Jia, Meng Wang, Yucheng Zhou

**Affiliations:** 1School of Information and Electrical Engineering, Shandong Jianzhu University, Jinan 250101, China; jmybl2016@163.com; 2National Supercomputer Research Center of Advanced Materials, Advanced Materials Institute, Qilu University of Technology (Shandong Academy of Sciences), Jinan 250014, China

**Keywords:** carbonized cotton fiber, PDMS, sensor, electrodes

## Abstract

Flexible pressure sensors with high sensitivity have great potential applications in wearable electronics. However, it is still a great challenge to prepare sense electrodes with high flexibility, high sensitivity, and high electrochemical performance. Here, we propose a novel and simple method for carbonizing cotton fibers as excellent electrically conductive materials. Moreover, carbonized cotton fiber (CCF) and polydimethylsiloxane (PDMS) were assembled into a flexible sense electrode. The CCF/PDMS electrode shows a high sensitivity of 10.8 kPa^−1^, a wide response frequency from 0.2–2.0 Hz, and durability over 900 cycles. The combined CCF/PDMS sensors can monitor human movement and pulse vibration, showing the enormous potential for use in wearable device technology. Additionally, the CCF/PDMS can be used as electrodes with a specific capacitance of 332.5 mF cm^−2^ at a current density of 5 mA cm^−2^, thanks to their high electrical conductivity and hydrophilicity, demonstrating the promising prospect of flexible supercapacitors.

## 1. Introduction

Flexible, sensitive, and low-cost pressure sensors have attracted considerable interest for their great potential applications in wearable electronics and intelligent systems [1,2]. Flexible pressure sensors, as a significant subfield of wearable electronics [3], present substantive potential applications in smart human-machine interaction [4], human structural health detection [5], and sports performance monitoring [6]. Recently, many reported studies have focused on four types of representative pressure sensors, which are based on transistor [7,8,9], piezoelectric [10,11,12], capacitive [13,14,15], and piezoresistive [16,17,18] mechanisms. Among these various sensing types, piezoresistive pressure sensors have attracted increasing attention for their excellent signal-collection capability, straightforward architecture, and manufacturing procedure [19,20,21,22]. Remarkably, it has been shown that patterning the surface of the sensing layer with microstructures is a successful method for obtaining a high sensitivity and a low detection limit when making piezoresistive sensors [8,23,24,25,26]. Polydimethylsiloxane (PDMS), with its hyperelastic characteristics, may be advantageous for usage in various contexts, such as lab-on-a-chip and micro- and nano-electromechanical systems (MEMS/NEMS) [27,28]. For instance, a capacitive pressure sensor based on microstructured PDMS sheets had sensitivity in the same pressure range that was around 30 times greater than the unstructured one [8]. Despite the fact that pure PDMS has a high deformation capacity, fillers or reinforcements are frequently added to produce composite materials that typically have a higher stiffness modulus, fracture toughness, fatigue resistance, tensile strength, and abrasion resistance [29,30].

Although conventional pressure-sensing platforms based on metal-wires and semiconductors have attracted extensive investigation, their rigidity, fragility, low resolution, and low sensing range limit their applications to wearable pressure sensors [31,32]. In addition, various flexible and wearable pressure sensors that are based on nanomaterials have been successfully developed. Due to their superior electrical and mechanical qualities, nanomaterials, such as metal nanoparticles, gold nanowire [33,34], and low-dimensional carbons (e.g., carbon nanotubes (CNTs) [35,36], carbon black [37], and graphene [38,39]), have been extensively used in the production of pressure sensors. Nonetheless, the fabrication process of these sensors is generally costly and complicated and may involve unknown biotoxicity of nanomaterials, which limits their applications in monitoring human motions. Therefore, a replacement with biocompatible, biodegradable, eco-friendly, and low-cost properties is urgently needed.

As the most abundant biopolymer on earth, cellulose is ubiquitous in our daily lives as a biodegradable and biocompatible template for the synthesis of materials with tailored functionality [40]. The amount of cellulose in cotton fiber, which may reach 95%, is the greatest of all lignocellulosic resources. Biomass is now being used to make carbon-based products since it is affordable, simple to get, sustainable, and ecologically beneficial [41]. Porous carbon from biomass, such as bacterial cellulose, raw cotton, and lignin, has shown a variety of potential applications in solar energy conversion and storage, super-capacitors, electromagnetic interference shielding, batteries, and water treatment due to their high porosity, flexibility, hydrophobicity, and electrical conductivity [42]. Cotton fiber has been widely used in the textile industry due to its fiber length advantage. Compared to wood cellulose, cotton has a highly crystalline cellulose component [43]. Despite their superior mechanical and electrical properties, carbon materials from wood cellulose have not yet been utilized in designing flexible sensors.

Herein, a straightforward and inexpensive method for creating an extremely sensitive pressure sensor based on cellulose is described, using carbonized cotton fiber (CCF) as the conductive filler and polydimethylsiloxane (PDMS) as the polymer matrix. The results demonstrate that the CCF/PDMS composite shows outstanding mechanical properties and electrochemical characteristics, which may be useful in the development of compressible supercapacitors and pressure sensors. This work offers an effective and straightforward method for creating a compressible and conductive CCF/PDMS composite, demonstrating the enormous potential for industrialization and mass production.

## 2. Experimental

### 2.1. Materials

The cotton fiber was kindly provided by Xinjiang Hongruida Fiber Co., Ltd. (Bazhou, China). Toluene and PDMS (Slygard 184, Dow Corning, Midland, MI, USA) were purchased from Beijing LanYi Chem Co., Ltd. (Beijing, China) and utilized directly.

### 2.2. Preparation of CCF/PDMS Pressure Sensor

Figure 1a illustrates the fabrication process of CCF/PDMS pressure sensors. Deionized water was used to rinse the cotton fiber to get rid of any solid impurities. The cotton textiles were preserved for later use in a dry place after being properly dried in a tube furnace with an environment of inert gas nitrogen (gas flow, 200 sccm purity, 99.999%). The cotton fiber was heated to 900 °C at a rate of 5 °C min^−1^, maintained at that temperature for 120 min, and then cooled to room temperature within 180 min. After that, the large amounts of carbonized cotton fiber were crushed and sieved, and the powder (CCF) passed through 200 mesh was gathered and employed as a conductive filler. A simple vacuum infusion procedure was used to create the CCF composite materials. Initially, the PDMS solution was created by thoroughly combining 2 g of the main agent (Sylgard 184, Dow Corning) and 0.2 g of the curing agent. The mixture was then diluted with 6 mL of toluene. The liquid was then violently stirred for 30 min while 1 g of carbonized cotton fiber was added, creating a homogeneous suspension. The suspension was then put into a PTFE container and left in a vacuum chamber at room temperature for 2 h to remove the bubbles and solvent. The sample was then cured for 3 h at 125 °C to create the CCF/PDMS composites.

### 2.3. Characterization

The morphologies and structures of materials were characterized by a field emission SEM (FE-SEM, FEI Quanta 650). A digital camera was used to take all of the given optical images (Canon IXUS 70). Using a 532 nm laser at room temperature, a laser Raman spectrometer (HPRIBA Evolution) examined the Raman spectra. The FTIR spectra of carbonized cotton fiber were obtained using a PerkinElmer infrared spectrometer (Spectrum 100 with) the method of potassium bromide pellet. An elemental analyzer (CE−440, PerkinElmer, Waltham, MA, USA) was used to measure the content of elements in CF and CCF. The resistance of the CCF/PDMS composite was evaluated at room temperature using a digital multimeter. The electrical conductivity of the CCF and CCF/PDMS composites was measured at room temperature with a two-probe method using an insulation resistance meter and a digital multimeter, respectively.

### 2.4. Electrochemical Measurements

The electrochemical workstation (CHI 660D, Chenhua, Shanghai, China) and microforce tester (UTM6503, SUNS CATALOG) were used to examine the pressure-sensing capabilities of the CCF/PDMS composite, which had a sample of 10 × 10 × 2 mm. The electrochemical workstation was used to record the electric current under 1.0 V on the composite. The relative change of resistance (RCR) was computed using the following formula: Δ*R/R*_0_ *= (R_p_ − R*_0_*)/R*_0_, where *R*_0_ and *R_p_* represent resistance without and with applied stress, respectively.

The electrochemical performances of CCF/PDMS composites were measured using a CHI 660D work station (Shanghai Chenhua Instruments Co., Shanghai, China) in a three-electrode configuration. The working electrode was made by sandwiching a patch of CCF/PDMS between two pieces of stainless-steel net without the use of any polymer binders or additives. The reference and counter electrodes were a saturated calomel electrode (SCE) and a Pt mesh, respectively. At room temperature, electrochemical experiments were conducted in a 1 M Na_2_SO_4_ aqueous solution. Cyclic voltammetry (*CV*) and galvanostatic charge/discharge (*GCD*) experiments were performed in the 0 to 1.0 V potential range. The electrochemical impedance spectroscopy (EIS) curves were measured at the open circuit potential with a frequency range of 100 kHz to 10 mHz and an amplitude of 5 mV. Moreover, the relevant specific capacitances were determined using Equations (1) and (2).
(1)$$C_{\mathrm{GCD}}=\frac{I\;\Delta t}{m\;\Delta v}$$
where *C_GCD_* (F/g) is the specific capacitance estimated from GCD curves, *I* (A) denotes instantaneous current, Δ*t* (s) denotes the discharge tie, *m* (g) denotes the mass on the SSF, and Δ*v* (*V*) denotes the potential window in the curve.
(2)$$C_{\mathrm{CV}}=\frac{\int I\;\left(V\right)dV}{m\;v\;\Delta V}$$
where *C_CV_* (F/g) is the specific capacitance computed from *CV* curves, *∫I (V)dV* is the current integrated inside the potential window at a certain scan rate, *m* (g) is the mass on the SSF, *v* (mV/s) is the scan rate, and Δ*V* (V) is the potential window in the curve.

## 3. Results and Discussion

The cotton fiber is mainly composed of cellulose. The chemical structure of cellulose is illustrated in Figure 1b. Cellulose is composed of β-d-glucopyranose units linked by 1, 4-glycosidic bonds [44]. The FESEM images of cotton fiber and carbonized cotton fiber are displayed in Figure 1c,d. As can be seen, the cotton fiber with a corrugated surface is composed of highly aligned and intimately bound fibers. The regular strip structure and alignment in the cotton fiber are well-arrayed with a size of 10–20 μm. After carbonization, Figure 1d exhibits the randomly oriented and loosely bound fibers in the CCF. It can be seen in Figure 1e that the conducive fillers (CCF) tightly fill in the non-conductive (PDMS) portions of the composite. Figure 1g depicts the Raman spectra of carbonized cotton fiber. It has been widely assumed that the Raman shift around 1360 cm^−1^ corresponds to the D peak, which represents defects or heteroatom doping, and that the band near 1580 cm^−1^ corresponds to the G peak, which represents the crystalline sp^2^ carbon. These two bands appear on the CCF. This confirms the formation of graphite-like carbon that can endow the CCF with good electrical conductivity. The *I_D_/I_G_* ratio is inversely related to the graphitization and conductivity of carbon materials [45]. According to the result of Gauss fitting of the Raman spectrum, the intensity ratio of *I_D_/I_G_* is 1.07, indicating that CCF has greater electrical conductivity.

The changes in carbonization-treated cotton fiber before and after treatment were analyzed by AIR-FTIR (Figure 2a,b). The characteristic peaks of cotton fiber at 3432 cm^−1^ and 1033 cm^−1^ were linked to the O-H stretching vibration and the C-O-C bond of cellulose, respectively [46]. The C-H stretching and bending of -CH_2_ groups were represented by the peaks at 2900 cm^−1^ and 1430 cm^−1^, respectively [47]. Moreover, the peak at 1630 cm^−1^ was attributed to the H-O-H stretching vibration of absorbed water in the carbohydrate. Compared with cotton fiber with CCF, the FTIR signals corresponding to the functional groups essentially vanished following carbonization. This is due to the fact that cotton fibers are composed of millions of cellulose molecules, and in the process of carbonization at high temperatures, cellulose undergoes dehydration, decarboxylation, and decarbonylation reactions, releasing H_2_O, CO_2_, CO, and other small molecules and transforming into carbon with a distorted graphite structure [48]. Figure 2c shows the XPS spectrum of cotton fiber and CCF. These peaks at around 285 eV and 531 eV are assigned to C 1s and O 1s, respectively. High-resolution XPS of C1s of cotton fibers and CCF and their deconvolution are shown in Figure 2d,e. The three characteristic peaks are located at 284.3 eV, 285.2 eV, and 288.3 eV, respectively, corresponding to C-C, C-O, and C=O bonds. It can be seen that the content of C-O and C=O in the CCF is significantly reduced, which is mainly due to the dehydration, decarboxylation, and decarbonylation reactions of cellulose during pyrolysis [48]. The elemental composition analysis of cotton fiber and CCF is shown in Table 1. After carbonization, the C content increases from 60.62% to 87.09%, and the O content decreases from 35.27% to 11.33%, which is consistent with the analysis results of the infrared spectrum. This indicates that there is a small amount of hydrogen and oxygen after the carbonization of cotton fiber. However, the CCF has good conductivity, indicating that the conductivity will not be weakened due to the absence of hydrogen and oxygen elements. Figure 2f shows the XRD spectra of cotton fiber and CCF. The XRD spectra of cotton fiber show three diffraction peaks at 14.5°, 16.7°, and 22.8° corresponding to the (1–10), (110), and (200) crystalline surfaces of cellulose, representing a typical cellulose I crystalline structure [49]. After carbonization, the characteristic peaks of cotton fiber almost disappear, and broad peaks of graphitic carbon appear around 23° and 44°, which proves that cotton fiber transforms into a graphitic carbon structure after carbonization [50].

As shown in Table 2, the conductivity of CCF is 12.21 S m^−1^, which is lower than that of graphene [51] and carbon nanotubes [52]. However, the preparation process of CCF is simple and low-cost, which is conducive to industrial production. The conductivity of the assembled CCF/PDMS composite film decreased to 0.43 S m^−1^. The conductivity of CCF/PDMS composite film mainly comes from conductive CCF. As shown in Figure 3, with the increase of carbonized cotton pulp fiber, the Δ*R/R*_0_ of CCF/PDMS composite film gradually increases. At the initial stage, the Δ*R/R*_0_ of the CCF/PDMS composite increases rapidly. When it increases to 35%, the Δ*R/R*_0_ growth of CCF/PDMS composite resistance has leveled off. As the CCF continues to increase, the content of PDMS decreases accordingly. As a result, the adhesion of PDMS becomes weak, which gradually reduces the flexibility of the CCF/PDMS composite film. Hence, 35% CCF is selected as the main research object of the following experiment.

The response of the CCF/PDMS sensor to multiple compression-relaxation cycles was recorded and presented in Figure 4. The current-voltage (I–V) characteristic of the CCF/PDMS sensor was tested at a pressure of 0–50 kPa using a sweeping voltage from −3 to 3 V. As shown in Figure 4a, all CCF/PDMS I-V curves exhibit a linear response to continuous stress, indicating an ohmic behavior and constant resistance. The slope of the I-V curves increases proportionally with pressure, suggesting that resistance decreases with pressure. Figure 4b depicts the response of the relative change of resistance (RCR) to stress, which illustrates the pressure sensitivity of the CCF/PDMS sensor. It can be seen that the RCR response increases rapidly within the range of pressures less than 4.5 kPa. When the pressure applied to the composite is greater than 4.5 kPa, the increasing trend of the relative change value of resistance slows down. According to the sensitivity calculation formula, the sensitivity of CCF/PDMS composites in the range of 0–4.5 kPa pressure is 10.8 kPa^−1^. This is higher than the previously reported which is higher than the previously reported carbon aerogel flexible sensor (0.26 kPa^−1^) [48], the carbonated fiber/PDMS flexible sensor (8.4 kPa^−1^) [53], and the carbon nanotube sensor (4.3 kPa^−1^) [19]. This is due to the unique surface profile of CCF particles, and when subjected to stress loads, the profile between conductive particles in the conductive network on the deformation of the elastic substrate favors the connection between conductive particles. In addition, the elastic substrate may simply move on the surface of the CCF without breaking or changing the distance between them, so that there is a chance for the connections between the two linked conductive fibers to be retained. Meanwhile, the preparation cost of CCF/PDMS composites is much lower than that of carbon nanomaterials, which has expanded the application of renewable cellulose resources. In Figure 4c, there is a strong correlation between the strength of the responding signal and the pressure input, which is a useful characteristic for pressure load detection. In Figure 4d, the response of the CCF/PDMS composite during multiple cycles of compression and relaxation is contrasted. The response behavior of the sensor is consistent between 0.2–2 Hz, indicating a quick and reliable response. In addition, it can be seen that the amplitude of RCR at high frequency is much larger than that at low frequency, which is due to the fact that at high frequency, a greater amount of stress is placed on the sensor with fixed strain than at low frequency. Figure 4 depicts the response of the CCF/PDMS sensor to multiple compression-relaxation cycles. As seen in Figure 4e, there is little drift and hysteresis between the loading compression wave and the RCR response of the sensor. Repeatable behavior is significantly important for the application of sensors. Under a pressure of 10 kPa, the robustness of the CCF/PDMS pressure sensor was tested (Figure 4f). After nearly 900 cycles, the intensity of the response of the CCF/PDMS composite remained strong despite repeated compression, suggesting good resilience of the CCF/PDMS composite to pressure input.

Due to its great chemical stability and adjustable mechanical characteristics, PDMS, a silicon-based elastomer with a repeating unit of SiO(CH_3_)_2_, is one of the most often utilized polymers for flexible devices. By vacuum infusing PDMS into the CCF scaffold, CCF/PDMS composites were created, taking advantage of the high porosity of CCF. The CCF/PDMS composite keeps the same dimensions without any apparent voids after curing, proving that the carbon fiber linkages withstood the infusion process. The CCF/PDMS composite displays outstanding bendability, as seen in Figure 5a. The conducting routes and contact resistance between the close-by conducive fillers have an impact on the change in resistance for a piezoresistive sensor.

The tunneling principle should be followed, and the contact resistance and conducive route between the nearly filled particles should determine how the resistance changes. According to the theory, the tunneling resistance in the CCF/PDMS composite can be expressed as [54]:(3)$$R_{tunnel}=\frac{V}{aJ}=\frac{2}{3}\frac{h^{2}s}{ae^{2}\sqrt{2m\varphi }}exp\left(\frac{4\pi }{h}\sqrt{2m\varphi }\;s\right)$$
where *s* is the distance between CLs in the insulating material, *h* is Planck’s constant, *e* is the quantum of electricity, *m* is the mass of an electron, *J* is the tunneling current density, *a* is the cross-sectional area of the tunnel, *V* is the potential difference between the conductive particle, and *φ* is the height of the potential barrier. According to Equation (3), when pressure was applied on the CCF/PDMS sensor (Figure 5c), the distance between the CCF decreased (a decrease of *s* in the formula), leading to the decrease of the *R_tunnel_*. The insulator PDMS may be readily distorted in the first stage with the pressure applied since it is an elastic matrix. Meanwhile, the conductive fibers suddenly contact each other, which causes a rapid drop in resistance and high sensitivity of the CCF/PDMS composite in the initial stage. This is consistent with the result in Figure 5b. However, with the increase in pressure, the deformation of the CCF/PDMS composite gradually increases. PDMS will resist deforming, and the decrease in distance between the CCF will be limited. Therefore, the pressure sensitivity is restricted in the high stress phase with a high RCR response.

A cotton fiber-based sensor possesses good flexibility, high sensitivity, and a wide range of strain gauges, so it was used to explore potential applications for human activities. In this study, a wearable device made of CCF/PDMS composites was used to assess pulse rate and strength. As revealed in Figure 6a, a flake-like CCF/PDMS sensor prototype was attached to the temples to track the human motion of blinking the eyes. It can be observed that the sensor responded well to the motions of the eyes. Repeated pulse rate signals may be seen while the eyes blink. This shows the exceptional ability to detect even the most subtle changes in the human body. We attached a CCF/PDMS sensor to an index finger and tested its reaction to finger bending (Figure 6b). When the extended finger was bent to a specific angle, the relative resistance changes of the strain sensor increased by a given amount and then stabilized. The resistance decreases during the straightening of the finger, thus forming a stepped signal. Similar measurements were conducted on an elbow joint with continual bending, and the strain sensor also showed good performance (Figure 6c). It can be seen that the sensor switched rapidly at loading and unloading, where the current value remained nearly the same under the same motion. This demonstrated the ability of the CCF/PDMS sensor to maintain and monitor large deformations of the human body. The relative curves are different and distinguishable by comparing the shape and intensity change of the plot. As shown in Figure 6d, weights of different masses (5 g, 10 g, 20 g, and 50 g) were placed on the surface of the strain sensor to detect the change in pulse rate. The response to the force on the CCF/PDMS sensor may be used to recognize the weight of weights. It is seen that the weight of weights can be recognized by the RCR response to the force used for loading them. The response signal grows in proportion to the weight of the weights. In addition, muscular action was recorded when speaking. A volunteer was asked to read different words, such as “Hi, Hello and Cellulose”. For each syllable, the CCF/PDMS strain sensor displayed a distinct signal pattern. This accurate and sensitive capability of the CCF/PDMS strain sensor has major applications in smart skin electronics, personalized health detection, and human-machine interaction.

### CCF/PDMS as Electrodes for Supercapacitors

The GCD, CV, and EIS measurements were performed in a three-electrode arrangement to investigate the electrochemical performance of CCF/PDMS. As revealed in Figure 7a, the GCD curves of the CCF/PDMS maintain an approximately rectangular form, and an internal resistance (IR) drop was detected, which is due to the internal resistance of the CCF/PDMS. The CV curves basically remain unchanged at various scan rates from 5 to 100 mV s^−1^, indicating that the electrode can maintain good charging and discharging behavior. Figure 7b shows the GCD curves of CCF/PDMS at different current densities. It can be seen that all curves show triangular and linear shapes, implying typical capacitance characteristics of CCF/PDMS. Moreover, according to Equation (1), when the current density is 1, 2, 3, 5, 7, and 10 mA cm^−2^, the specific capacitance of CCF/PDMS is 559.9, 468.8, 398.4, 332.5, 283.5, and 223 mF cm^−2^, respectively (Figure 7c). The mass specific capacitance of CCF/PDMS is 89.3 F g^−1^ at a current density of 1 mA cm^−2^. In order to study the application of CCF/PDMS composites in flexible electrodes, the electrochemical performance of CCF/PDMS under different strains was tested. As shown in Figure 7d, the CV curves under different strains present approximately rectangular shapes at a scan rate of 25 mV s^−1^, demonstrating that CCF/PDMS composites can maintain the capacitance behavior of EDLC under different strains. Figure 7e shows the GCD curves of CCF/PDMS composites at a current density of 3 mA cm^−2^. The initial capacitance retention of the CCF/PDMS composites at different strains (0%, 45%, and 90%) was about 98.3%, 97.6%, and 96.5%. With the increase in strain, the internal resistance of CCF/PDMS decreases accordingly. To further investigate the electrochemical transfer mechanism of the CCF/PDMS, the EIS of the CCF/PDMS was conducted from 0.1 to 100 kHz under various stresses (Figure 7f). A straight line and a conventional semicircle appear in all of the Nyquist plots. Fitting the impedance data with the equivalent circuit was used to examine the measured impedance spectra. The equivalent series resistance (ESR) fell from 15.3 to 11.4, signifying improved charge-transfer capacity at the electrode/electrolyte interfaces and lower ion diffusion resistance. This is due to increased conductivity and a significantly shorter ion diffusion route at higher stresses.

## 4. Conclusions

In conclusion, a facile and simple approach is proposed to fabricate the flexible, conductive, and pressure-sensitive composite material with carbonized cotton fiber and polydimethylsiloxane (CCF/PDMS). A simple carbonization procedure was devised to effectively remove oxygen and hydrogen from cotton fiber. The obtained conductivity of CCF is 12.21 S m^−1^, and the conductivity of the CCF/PDMS composite is 0.43 S m^−1^, with a CCF concentration of 35%. The CCF/PDMS composite shows a high sensitivity of 10.8 kPa^−1^, a wide response frequency from 0.2–2.0 Hz, and durability over 900 cycles. The assembled sensor might be used to detect human activities such as facial expression, muscle movement, and force requirements. Furthermore, the CCF/PDMS can be used as electrodes with a specific capacitance of 332.5 mF cm^−2^ at a current density of 5 mA cm^−2^. It is worth noting that the device fabrication process is simple and advantageous for large-scale production with low-cost cotton fiber as a raw material. The CCF/PDMS composites are believed to have promising potential applications in sensors and supercapacitors.

## Figures and Tables

**Figure 1 polymers-15-02095-f001:**
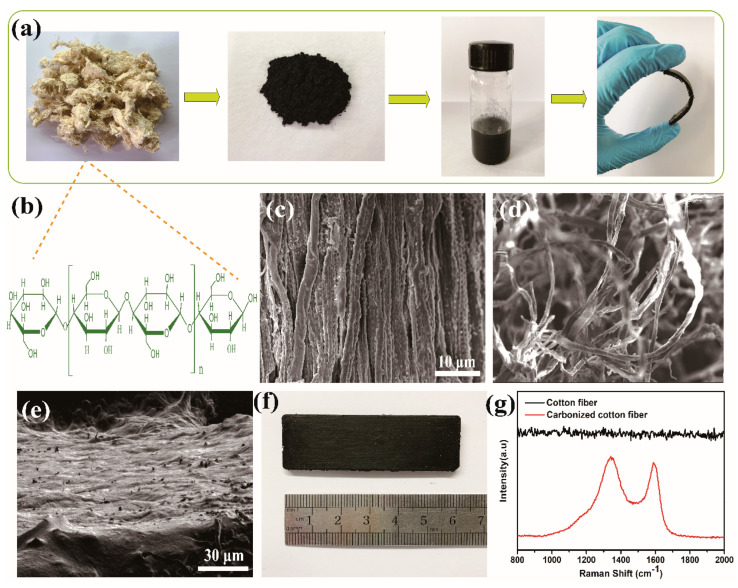
(**a**) Schematic of the fabrication of CCF/PDMS; (**b**) The chemical structure of cellulose; (**c**,**d**) FESEM images of cotton fiber and CCF; (**e**) An FESEM image showing the cross-sectional morphology of the CCF/PDMS; (**f**) Photograph showing the CCF/PDMS; (**g**) Raman spectra of cotton fiber and CCF.

**Figure 2 polymers-15-02095-f002:**
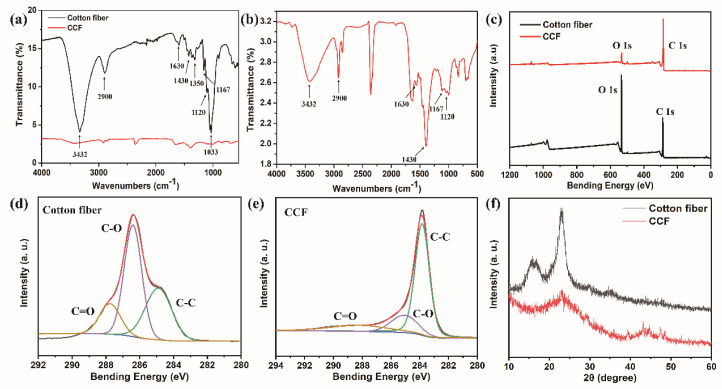
(**a**) FTIR spectra of cotton fiber and CCF; (**b**) the amplified spectrum of CCF; (**c**) XPS spectra of cotton fiber and CCF; (**d**,**e**) high-resolution XPS spectra of C 1 s of cotton fiber and CCF; (**f**) XRD spectra of cotton fiber and CCF.

**Figure 3 polymers-15-02095-f003:**
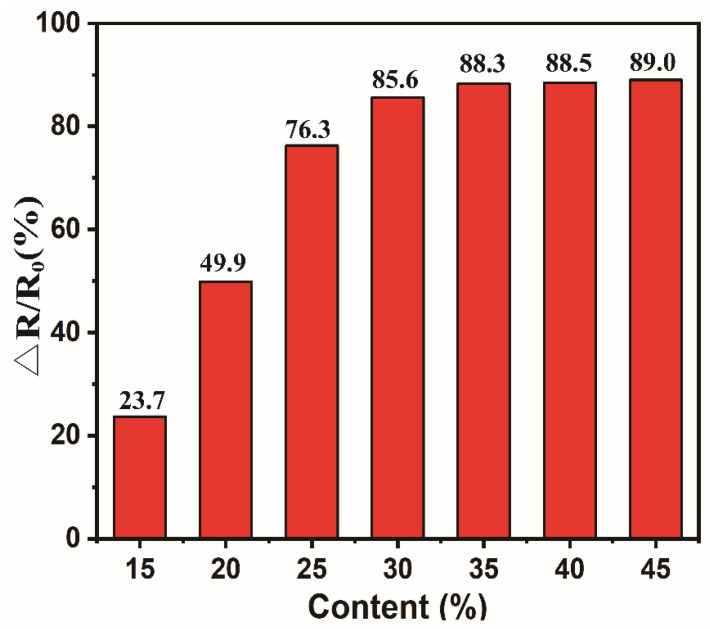
Relative change of resistance of CCF/PDMS composite with different contents of CCF.

**Figure 4 polymers-15-02095-f004:**
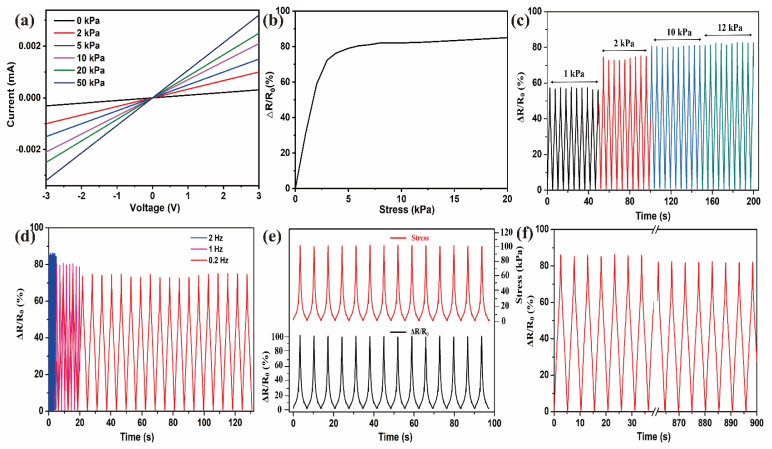
(**a**) Current-voltage curves of CCF/PDMS composite under different pressures; (**b**) Pressure response curves for CCF/PDMS; (**c**) Response of CCF/PDMS to different loading cycles; (**d**) Response of CCF/PDMS at different frequencies (100 kPa, 15 times); (**e**) Response of the CCF/PDMS sensor under repeated compression-relaxation cycles; (**f**) Reliability test of the CCF/PDMS sensor under repeated compression-relaxation cycles in the range of 0–100 kPa.

**Figure 5 polymers-15-02095-f005:**
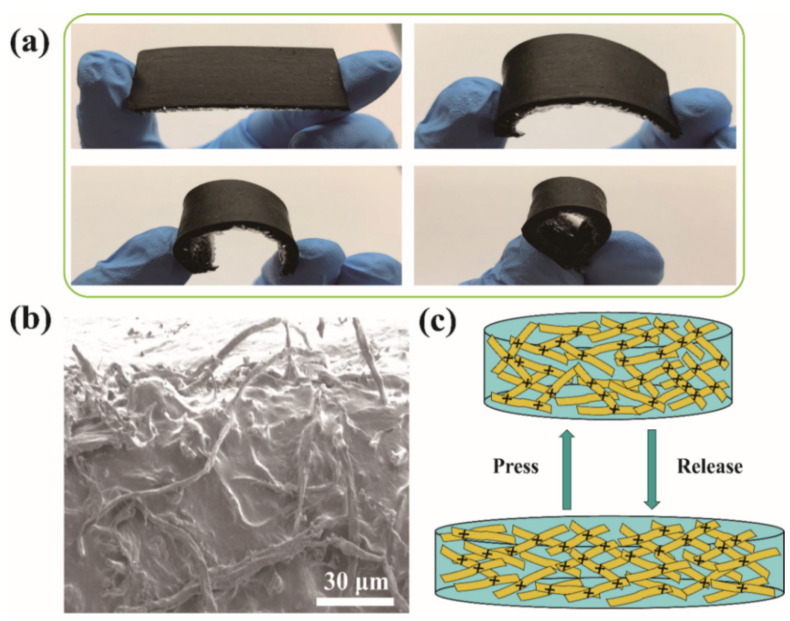
(**a**)The picture of deformation of CCF/PDMS composite. (**b**) Fracture surface of the CCF/PDMS composite; (**c**) Diagram of the inner structural change in the CCF/PDMS composite without and with applied stress; the carbonized cotton fiber is represented by a yellow rectangle, and the CCF connection point is shown by the black cross.

**Figure 6 polymers-15-02095-f006:**
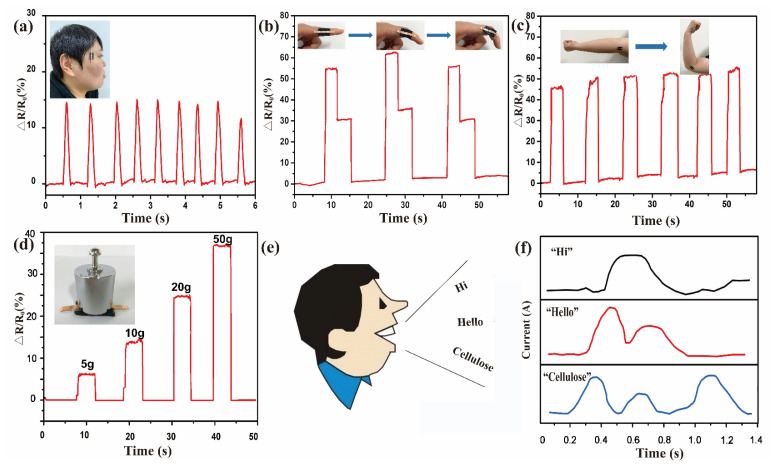
The CCF/PDMS sensor was used to monitor human activities: (**a**) Eye blinking; (**b**) Finger bending; (**c**) bending-release movement of the elbow; (**d**) Measurement of power for loading different weights (5 g, 10 g, 20 g, and 50 g); (**e**,**f**) Measurement of different sound stimuli.

**Figure 7 polymers-15-02095-f007:**
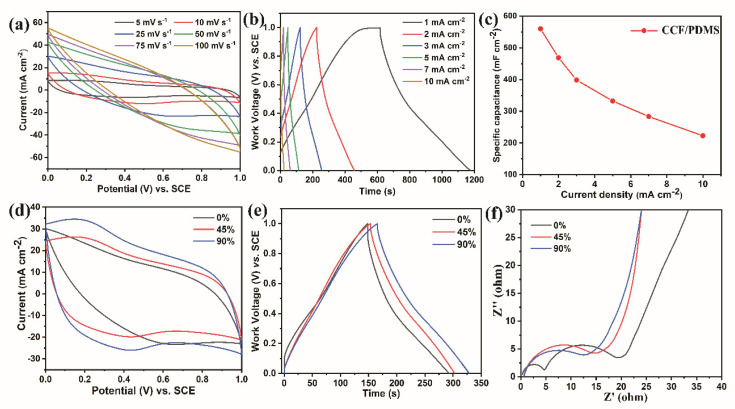
(**a**,**b**) CV and GCD curves of the CCF/PDMS composite film; (**c**) specific capacitance of CCF/PDMS composite film at various current densities; (**d**,**e**) CV and CD curves of CCF/PDMS under different strains; (**f**) EIS spectra of CCF/PDMS under different strains.

**Table 1 polymers-15-02095-t001:** Elemental analysis of CF and CCF.

Sample	C (%)	H (%)	O (%)	N (%)
CF	60.62	3.53	35.27	0.58
CCF	87.09	1.21	11.33	0.37
Changes (%)	43.67	−65.72	−67.88	−36.21

**Table 2 polymers-15-02095-t002:** Electrical conductivities of CCF and CCF/PDMS composites.

Electrical Conductivity	CF	Graphene	CNT	CCF	PDMS	CCF/PDMS
Average (S m^−1^)	0	6.6 × 10^4^ [51]	5.15 S cm^−1^ [52]	12.21	1.25 × 10^−11^	0.43
Standard Deviation (S m^−1^)	0	/	/	2.03	1.10 × 10^−12^	0.12

## Data Availability

Data are contained within the article.

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
