# Peer review of "A Flexible and Highly Sensitive Pressure Sense Electrode Based on Cotton Pulp for Wearable Electronics"

_polymers, 2023, doi:10.3390/polym15092095_

Round 1

Reviewer 1 Report

1.       In the abstract, the composite shows durability over 900 cycles of what?

2.       In equation 1, is lower Δt is it correct, or is it supposed to be ΔV? And Δt(s) denotes the discharge time (not tie)

3.       The statement about “cellulose being composed of β-D-glucopyranose units..” should be backed with a reference.

4.       In the Figure 2 caption, there is no description of figures (e) and (f)

5.       Fig. 5d and 5e in pg. 5 are referring to which figures?

6.       Fig. 5f – which figure?

7.       Table 2 – Electrical conductivity of cotton fibers can be included to justify carbonization. Also, the conductivity of graphene and carbon nanotubes can be included plus its references. This will bring out the comparison of conductivity.

8.       Include references for the tunnelling principle

9.       The number of the equation on pg. 8, needs to be revised because it’s not equation 1

10.   Fig 6d, lists the weights mentioned for the different masses used

11.   Include references from the last five years

Reviewer 2 Report

In this work, the authors developed a flexible and highly sensitive pressure-sense electrode based on the cotton pulp for wearable electronics. The research appears to be efficiently done and appropriately reported, however, the standard of English must be improved and revised by a native English expert. Nevertheless, some questions and corrections must be answered to improve and complete the manuscript.

Abstract section: The abstract must be improved; I suggest to authors follow these rules:

A. One or two sentences on BACKGROUND

B. Two or three sentences on METHODS

C. Less than two sentences on RESULTS

D. One sentence on CONCLUSIONS

Introduction section: In this section, the authors do not indicate the novelty of their work. what is the innovation of your work when compared with the other researchers? The "Knowledge gap to be filled"? In this introduction, the authors must describe or indicate the work that will be done to test their "hypothesis". The references must be updated. On the other hand, the authors do not analyze the mechanical properties of the PDMS, in particular, its hyperelasticity, which allows the high flexibility of the sensor developed in this work. Therefore, I suggest that authors consult new references on this subject, namely, the works described in the references whose DOI is 10.3934/matersci.2019.1.9 and 10.3390/polym13234258.

Experimental section. In my opinion, the work could be improved if the authors implemented some mechanical characterization, namely, tensile tests of the composite CCF/PDMS to evaluate its flexibility when compared with pure PDMS. Another mechanical test that I suggest is the hardness measurement using the Shore scale.

Change the position of equation 2.

Figures 2, 4, and 7: Poor quality images. The authors must improve the resolution of these figures.

Page 5, line 3. Please, change “… following carbonization..” to the following carbonization..”.

Page 6, second paragraph. The authors indicated that the sensor was tested at a pressure of 0-200KPa. Why this range?

Page 6, second paragraph, line 7. At first time that the authors used an abbreviation they must indicate its meaning, so I call the attention to the RCR abbreviation.

Page 6, second paragraph, lines 12 and 13. Please, change “kpa” to “kPa”.

Page 6, second paragraph, line 9. Please, explain the sentence “… When the resistance is greater than 4.5 kPa, …”. How can a resistance be measured in kPa?

Page 8, third paragraph, line 4. Please, change “According to formula (1),…” to “According to equation (1),...”.

Reviewer 3 Report

This manuscript reports a highly flexible and good electrical conductive sense electrode based on sustainable biomass-derived materials which is fabricated through a facile, low-cost, and scalable approach. The authors used several characterization methods to test the composite materials which include SEM, CV, EIS, XPS, and FTIR analysis.

The authors can improve the manuscript by clarifying some of the characterizations used. For example, it is not clear how the conductivity is measured in the Table 2. While a simple digital multimeter can be used to measure a resistance, a conductance must be carefully measured. I have listed those concerns below,

1.      Pressure sensor characterization

a.      Muscular action recording with the sensor? More description is required for this measurement. Does the sensor is attached to the neck to monitor the muscular action?

b.      Equation (1): is the s inside the square root or outside?

c.      The authors mentioned that the resistance change rate of the sensor decreases as the pressure increases (the paragraph above Figure 6). How do you support this statement? Do you conduct any experiment to show this?

d.      Please specify the weights used in Fig. 6d. How is the resistance change related to weight increase?

2.      What does the SD mean in the Table 2? If this refers to a standard deviation, I suggest you use a standard deviation instead of SD.

3.      (page 5, line #14-15) Where is the Table 5.3?

4.      (page 5, line #19) Where is Fig. 5f?

5.      (page 5, line #10) Where are Fig. 5d and 5e?

6.      Figure 3 caption should not be separated from the Figure 3 itself.

7.      Figure 1 caption includes typo such as double semi colons.

8.      How did you measure the conductivity shown in the Table 2?

9.      How did you measure the normalized resistance change (ΔR/R) in Figure 3? Did you use a digital multimeter? What is the meanings of ΔR here?

(page 4 (line # 8), page 8(line #4, 5, 16)) conducive fillers à conductive fillers?

Reviewer 4 Report

Dear authors,

thank you for the article. Here are some corrections and improvements: 

You write it is an environmentally friendly process, but large amounts of toluene are used in the production. This is not environmentally friendly, isn't it?

Figure caption 2 ist not correct. 

What is the purpose of the Raman spectrum? It is not surprising that a carbon peak appears after carbonization. The chemical structure has changed, which is not surprising?

After "carbonization" only one point "." on page 5.

It is also not surprising that the XPS spectrum is changing. What conclusions are drawn from this? Would a comparison with carbon black or nanotubes make sense?

You can use directly carbon black instead of CCF!

I doubt that this is a quantum mechanical tunnel effect! Rather, the carbon particles come into contact with each other through pressing. Why is referenced here to a quantum mechanical effect in the nanometer range?

It is also very striking that almost exclusively Chinese/Asian authors are used in the references! International literature should also be quoted for an international paper.

Round 2

Reviewer 2 Report

The second version of the manuscript improved significantly when compared with the first version. So, in my opinion, the manuscript can be accepted for publication

Reviewer 4 Report

Dear authors,

thank you for the improvements and corrections in the article.

Best wishes from Germany

Michael